# Learning From Weakly Supervised Data by The Expectation Loss SVM (e-SVM) algorithm

**Jun Zhu**
Department of Statistics
University of California, Los Angeles
jzh@ucla.edu

**Junhua Mao**
Department of Statistics
University of California, Los Angeles
mjhustc@ucla.edu

**Alan Yuille**
Department of Statistics
University of California, Los Angeles
yuille@stat.ucla.edu

## Abstract

In many situations we have some measurement of confidence on "positiveness" for a binary label. The "positiveness" is a continuous value whose range is a bounded interval. It quantifies the affiliation of each training data to the positive class. We propose a novel learning algorithm called *expectation loss SVM* (e-SVM) that is devoted to the problems where only the "positiveness" instead of a binary label of each training sample is available. Our e-SVM algorithm can also be readily extended to learn segment classifiers under weak supervision where the exact positiveness value of each training example is unobserved. In experiments, we show that the e-SVM algorithm can effectively address the segment proposal classification task under both strong supervision (e.g. the pixel-level annotations are available) and the weak supervision (e.g. only bounding-box annotations are available), and outperforms the alternative approaches. Besides, we further validate this method on two major tasks of computer vision: semantic segmentation and object detection. Our method achieves the state-of-the-art object detection performance on PASCAL VOC 2007 dataset.

## 1   Introduction

Recent work in computer vision relies heavily on manually labeled datasets to achieve satisfactory performance. However, the detailed hand-labelling of datasets is expensive and impractical for large datasets such as ImageNet [6]. It is better to have learning algorithms that can work with data that has only been weakly labelled, for example by putting a bounding box around an object instead of segmenting it or parsing it into parts.

In this paper we present a learning algorithm called expectation loss SVM (e-SVM). It requires a method that can generate a set of proposals for the true label (e.g., the exact silhouette of the object). But this set of proposals may be very large, each proposal may be only partially correct (the correctness can be quantified by a continues value between $0$ and $1$ called "positiveness"), and several proposals may be required to obtain the correct label. In the training stage, our algorithm can deal with the strong supervised case where the positiveness of the proposals are observed, and can easily extend to the weakly supervised case by treating the positiveness as latent variables. In the testing stage, it will predict the label for each proposal and provide a confidence score.

There are some alternative approaches for this problem, such as Support Vector Classification (SVC) and Support Vector Regression (SVR). For the SVC algorithm, because this is not a standard binary

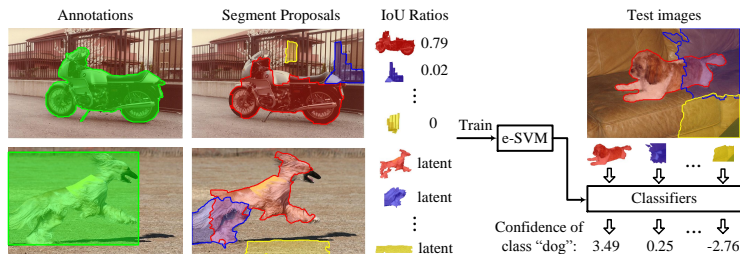

Figure 1: The illustration of our algorithm. In the training process, the e-SVM model can handle two types of annotations: pixel level (strong supervision) and bounding box (weak supervision) annotations. For pixel level annotations, we set the positiveness of the proposal as IoU overlap ratios with the groundtruth and train classifiers using basic e-SVM. For bounding box annotations, we treat the positiveness as latent variables and use latent e-SVM to train classifiers. In the testing process, the e-SVM will provide each segment proposal a class label and a confidence score. (Best viewed in color)

classification problem, one might need to binarize the positiveness using ad-hoc heuristics to determine a threshold, which degrades its performance [18]. To address this problem, previous works usually used SVR [4, 18] to train the class confidence prediction models in segmentic segmentation. However, it is also not a standard regression problem since the value of positiveness belongs to a bounded interval $[0, 1]$. We compare our e-SVM to these two related methods in the segment proposal confidence prediction problem. The positiveness of each segment proposal is set as the Intersection over Union (IoU) overlap ratio between the proposal and the pixel level instance groundtruth. We test our algorithm under two types of scenarios with different annotations: the pixel level annotations (positiveness is observed) and the bounding box annotations (positiveness is unobserved). Experiments show that our model outperforms SVC and SVR in both scenarios. Figure 1 illustrates the framework of our algorithm.

We further validate our approach on two fundamental computer vision tasks: (i) semantic segmentation, and (ii) object detection. Firstly, we consider semantic segmentation. There has recently been impressive progress at this task using rich appearance cues. Segments are extracted from images [1, 3, 4, 12], appearance cues are computed for each segment [5, 21, 25], and classifiers are trained using groundtruth pixel labeling [18]. Methods of this type are almost always among the winners of the PASCAL VOC segmentation challenge [5]. But all these methods rely on datasets which have been hand-labelled at the pixel level. For this application we generate the segment proposals using CPMC segments [4]. The positiveness of each proposal is set as the Intersection over Union (IoU) overlap ratio. We show that appearance cues learnt by e-SVM, using either the bounding box annotations or pixel level annotations, are more effective than those learnt with SVC and SVR on PASCAL VOC 2011 [9] segmentation dataset. Our algorithm is also flexible enough to utilize additional bounding box annotations to further improve the results.

Secondly, we address object detection by exploiting the effectiveness of segmentation cues and coupling them to existing object detection methods. For this application, the data is only weakly labeled because the groundtruth for object detection is typically specified by bounding boxes (e.g. PASCAL VOC [8, 9] and Imagenet [6]), which means that pixel level groundtruth is not available. We use either CPMC or super-pixels as methods for producing segment proposals. IoU is again used to represent the positiveness of the proposals. We test our approach on the PASCAL dataset using, as our base detector, the Regions with CNN features (RCNN) [14] (currently state of the art on PASCAL and outperforms previous works by a large margin). This method first used selective search method [24] to extract candidate bounding boxes. For each candidate bounding box, it extracts features by deep networks [16] learned on Imagenet dataset and fine-tuned on PASCAL. We couple our appearance cues to this system by simple concatenating our spatial confidence map features based on the trained e-SVM classifiers and the deep learning features, and then train a linear SVM. We show that this simple approach yields an average improvement of 1.5 percent on per-class average precision (AP).

We note that our approach is general. It can use any segment proposal detectors, any image features, and any classifiers. When applied to object detection it could use any base detector, and we could couple the appearance cues with the base detector in many different ways (we choose the simplest).

In addition, it can handle other classification problems where only the "positiveness" of the samples instead of binary labels are available.

## 2  Related work on weakly supervised learning and weighted SVMs

We have introduced some of the most relevant works published recently for semantic segmentation or object detection. In this section, we will briefly review related work of weakly supervised learning methods for segment classification, and discuss the connection to instance weighted SVM in literature.

The problem settings for most previous works generally assumed that they only get a set of accompanying words of an image or a set of image level labeling, which is different from the problem settings in this paper. Multiple Instance Learning (MIL) [7, 2] was adopted to solve these problems [20, 22]. MIL handles cases where at least one positive instance is present in a positive bag and only the labels of a set of bags are available. Vezhnevets *et.al.* [26] proposed a Multi-Image Model (MIM) to solve this problem and showed that MIL in [22] is a special case of MIM. Later, [26] developed MIM to a generalized MIM and used it as their segmentation model. Recently, Liu *et.al.* [19] presented a weakly-supervised dual clustering approach to handle this task.

Our weakly supervised problem setting is in the middle between these settings and the strong supervision case (i.e. the full pixel level annotations are available). It is also very important and useful because bounding box annotations of large-scale image dataset are already available (e.g. Imagenet [6]) while the pixel level annotations of large datasets are still hard to obtain. This weakly supervised problem cannot be solved by MIL. We cannot assume that at least one "completely" positive instance (i.e. a CPMC segment proposals) is present in a positive bag (i.e. a groundtruth instance) since most of the proposals will contain both foreground pixels and background pixels. We will show how our e-SVM and its latent extension address this problem in the next sections.

In machine learning literature, the weighted SVM (WSVM) methods [23, 27, **?**] also use an instance-dependent weight on the cost of each example, and can improve the robustness of model estimation [23], alleviate the effect of outliers [27], leverage privileged information [17] or deal with unbalanced classification problems. The difference between our e-SVM and WSVMs mainly lies in that it weights labels instead of data points, which leads to each example contributing both to the costs of positive and negative labels. Although the loss function of e-SVM model is different from those of WSVMs, it can be effortlessly solved by any standard SVM solver (e.g., LibLinear [10]) like those used in WSVMs. This is an advantage because it does not require a specific solver for the implementation of our e-SVM.

## 3  The expectation loss SVM model

In this section, we will first describe the basic formulation of our expectation loss SVM model (e-SVM) in section 3.1 when the positiveness of each segment proposal is observed. Then, in section 3.2, a latent e-SVM model is introduced to handle the weak supervision situation where the positiveness of each segment proposal is unobserved.

### 3.1  The basic e-SVM model

We are given a set of training images $\mathcal{D}$. Using some segmentation method (we adopt CPMC [4] in this work), we can generate a set of foreground segment proposals $\{S_1, S_2, \ldots, S_N\}$ from these images. For each segment $S_i$, we extract feature $\mathbf{x}_i$, $\mathbf{x}_i \in \mathbb{R}^d$.

Suppose the pixelwise annotations are available for all the groundtruth instances in $\mathcal{D}$. For each object class, we can calculate the IoU ratio $u_i$ ($u_i \in [0, 1]$) between each segment $S_i$ and the groundtruth instances labeling, and set the positiveness of $S_i$ as $u_i$ (although positiveness can be some functions of IoU ratio, for simplicity, we just set it as IoU and use $u_i$ to represent the positiveness in the following paragraphs). Because many foreground segments overlap partially with the groundtruth instances (i.e. $0 < u_i < 1$), it is not a standard binary classification problem for training. Of course, we can define a threshold $\tau_b$ and treat all the segments whose $u_i \geq \tau_b$ as positive examples and the segments whose $u_i < \tau_b$ as negative examples. In this way, this problem is transferred to a Support Vector Classification (SVC) problem. But it needs some heuristics to determine $\tau_b$ and its performance is only partially satisfactory [18].

To address this issue, we proposed our expectation loss SVM model as an extension of the classical SVC models. In this model, we treat the label $Y_i$ of each segment as an unobserved random variable. $Y_i \in \{-1, +1\}$. Given $\mathbf{x}_i$, we assume that $Y_i$ follows a Bernoulli distribution. The probability of $Y_i = 1$ given $\mathbf{x}_i$ (i.e. the success probability of the Bernoulli distribution) is denoted as $\mu_i$. We assume that $\mu_i$ is a function of the positiveness $u_i$, i.e. $\mu_i = g(u_i)$. In the experiment, we simply set $\mu_i = u_i$.

Similar to the traditional linear SVC problem, we adopt a linear function as the prediction function: $F(\mathbf{x}_i) = \mathbf{w}^T \mathbf{x}_i + b$. For simplicity, we denote $[\mathbf{w}\ b]$ as $\mathbf{w}$, $[\mathbf{x}_i\ 1]$ as $\mathbf{x}_i$ and $F(\mathbf{x}_i) = \mathbf{w}^T \mathbf{x}_i$ in the remaining part of the paper. The loss function of our e-SVM is the expectation over the random variables $Y_i$:

$$
\begin{aligned}
\mathcal{L}(\mathbf{w}) =& \lambda_{\mathbf{w}} \cdot \frac{1}{2} \mathbf{w}^T \mathbf{w} + \frac{1}{N} \sum_{i=1}^{N} \mathbb{E}_{Y_i}[\max(0, 1 - Y_i \mathbf{w}^T \mathbf{x}_i)] \\
=& \lambda_{\mathbf{w}} \cdot \frac{1}{2} \mathbf{w}^T \mathbf{w} + \frac{1}{N} \sum_{i=1}^{N} [l_i^+ \cdot \Pr(Y_i = +1|\mathbf{x}_i) + l_i^- \cdot \Pr(Y_i = -1|\mathbf{x}_i)] \qquad (1) \\
=& \lambda_{\mathbf{w}} \cdot \frac{1}{2} \mathbf{w}^T \mathbf{w} + \frac{1}{N} \sum_{i=1}^{N} \{l_i^+ \cdot g(u_i) + l_i^- \cdot [1 - g(u_i)]\}
\end{aligned}
$$

where $l_i^+ = \max(0, 1 - \mathbf{w}^T \mathbf{x}_i)$ and $l_i^- = \max(0, 1 + \mathbf{w}^T \mathbf{x}_i)$.

Given the pixelwise groundtruth annotations, $g(u_i)$ is known. From Equation 1, we can see that it is equivalent to "weight" each sample with a function of its positiveness. The standard linear SVM solver is used to solve this model with loss function of $\mathcal{L}(\mathbf{w})$. In the experiments, we show that the performance of our e-SVM is much better than SVC and slightly better than Support Vector Regression (SVR) in the segment classification task.

## 3.2 The latent e-SVM model

One of the advantage of our e-SVM model is that we can easily extend it to the situation where only bounding box annotations are available (this type of labeling is of most interest in the paper). Under this weakly supervised setting, we cannot obtain the exact value of the positiveness (IoU) $u_i$ for each segment. Instead, $u_i$ will be treated as a latent variable which will be determined by minimizing the following loss function:

$$
\mathcal{L}(\mathbf{w}, \mathbf{u}) = \lambda_{\mathbf{w}} \cdot \frac{1}{2} \mathbf{w}^T \mathbf{w} + \frac{1}{N} \sum_{i=1}^{N} \{l_i^+ \cdot g(u_i) + l_i^- \cdot [1 - g(u_i)]\} + \lambda_R \cdot R(\mathbf{u}) \qquad (2)
$$

where $\mathbf{u}$ denotes $\{u_i\}_{i=1,\dots,N}$. $R(\mathbf{u})$ is a regularization term for $\mathbf{u}$. We can see that the loss function in Equation 1 is a special case of that in Equation 2 by setting $\mathbf{u}$ as constant and $\lambda_R$ equal to 0.

When $\mathbf{u}$ is fixed, $\mathcal{L}(\mathbf{w}, \mathbf{u})$ is a standard linear SVM loss, which is convex with respect to $\mathbf{w}$. When $\mathbf{w}$ is fixed, $\mathcal{L}(\mathbf{w}, \mathbf{u})$ is also a convex function if $R(\mathbf{u})$ is a convex function with respect to $\mathbf{u}$. The IoU between a segment $S_i$ and groundtruth bounding boxes, denoted as $u_i^{bb}$, can serve as an initialization for $u_i$. We can iteratively fix $\mathbf{u}$ and $\mathbf{w}$, and solve the two convex optimization problems until it converges. The pseudo-code for the optimization algorithm is shown in Algorithm 1.

---

**Algorithm 1** The optimization for training latent e-SVM

**Initialization:**
  1: $\mathbf{u}^{(cur)} \leftarrow \mathbf{u}^{bb}$;
**Process:**
  2: **repeat**
  3:    $\mathbf{w}^{(new)} \leftarrow \arg\min_{\mathbf{w}} \mathcal{L}(\mathbf{w}, \mathbf{u}^{(cur)})$;
  4:    $\mathbf{u}^{(new)} \leftarrow \arg\min_{\mathbf{u}} \mathcal{L}(\mathbf{w}^{(new)}, \mathbf{u})$;
  5:    $\mathbf{u}^{(cur)} \leftarrow \mathbf{u}^{(new)}$;
  6: **until** Converge

---

If we do not add any regularization term on $\mathbf{u}$ (i.e. set $\lambda_R = 0$), $\mathbf{u}$ will become 0 or 1 in the optimization step in line 4 of algorithm 1 because the loss function becomes a linear function with respect to $\mathbf{u}$ when $\mathbf{w}$ is fixed. It turns to be similar to a latent SVM and can lead the algorithm to stuck in the local minimal as shown in the experiments. The regularization term will prevent this situation under the assumption that the true value of $\mathbf{u}$ should be around $\mathbf{u}^{bb}$.

There are a lot of different designs of the regularization term $R(\mathbf{u})$. In practice, we use the following one based on the cross entropy between two Bernoulli distributions with success probability $u_i^{bb}$ and $u_i$ respectively.

$$
\begin{aligned}
R(\mathbf{u}) &= -\frac{1}{N} \sum_{i=1}^{N} [u_i^{bb} \cdot \log(u_i) + (1 - u_i^{bb}) \cdot \log(1 - u_i)] \\
&= -\frac{1}{N} \sum_{i=1}^{N} D_{KL}[\text{Bern}(u_i^{bb})||\text{Bern}(u_i)] + C
\end{aligned}
\tag{3}
$$

where $C$ is a constant value with respect to $\mathbf{u}$. $D_{KL}(.)$ represents the KL distance between two Bernoulli distributions. This regularization term is a convex function with respect to $\mathbf{u}$ and achieves its minimal when $\mathbf{u} = \mathbf{u}^{bb}$. It is a strong regularization term since its value increases very fast when $\mathbf{u} \neq \mathbf{u}^{bb}$.

## 4 Visual Tasks

### 4.1 Semantic segmentation

We can easily apply our e-SVM model to the semantic segmentation task with the framework proposed by Carreira et al. [5]. Firstly, CPMC segment proposals [4] are generated and the second-order pooling features [5] are extracted from each segment. Then we train the segment classifiers using either e-SVM or latent e-SVM according to whether the groundtruth pixel-level annotations are available. In the testing stage, the CPMC segments are sorted based on their confidence scores output by the trained classifiers. The top ones will be selected to produce the predicted semantic label map.

### 4.2 Object detection

For the task of object detection, we can only acquire bounding-box annotations instead of pixel-level labeling. Therefore, it is natural to apply our latent e-SVM in this task to provide complementary information for the current object detection system.

In the state-of-the-art object detection systems [11, 13, 24, 14], the window candidates of foreground object are extracted from images and the confidence scores are predicted on them. Window candidates are extracted either by sliding window approaches (used in e.g. the deformable part-based model [11, 13]) or most recently, the Selective Search method [24] (used in e.g. the Region Convolutional Neural Networks [14]). This method lowers down the number of window candidates compared to the traditional sliding window approach.

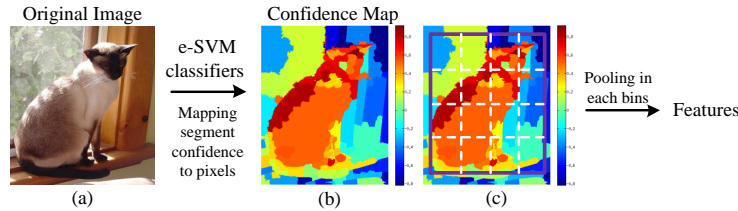

Figure 2: The illustration of our spatial confidence map features for window candidates based on e-SVM. The confidence scores of the segments are mapped to pixels to generate a pixel-level confidence map. We will divide a window candidate into $m \times m$ spatial bins and pool the confidence scores of the pixels in each bin. It leads to a $m \times m$ dimensional feature.

It is not easy to directly incorporate confidence scores of the segments into these object detection systems based on window candidates. The difficulty lies in two aspects. First, only some of the segments are totally inside a window candidate or totally outside the window candidate. It might be hard to calculate the contribution of the confidence score of a segment that only partially overlaps with a window candidate. Second, the window candidates (even the groundtruth bounding boxes) will contain some of the background regions. Some regions (e.g. the regions near the boundary of the window candidates) will have higher probability to be the background region than the regions in the center. Treating them equally will harm the accuracy of the whole detection system.

In order to solve these issues, we propose a new spatial confidence map feature. Given an image and a set of window candidates, we first calculate the confidence scores of all the segments in the image using the learned e-SVM models. The confidence score for a segment $S$ is denoted as $\mathrm{CfdScore}(S)$. For each pixel, the confidence score is set as the maximum confidence score of all the segments that contain this pixel. $\mathrm{CfdScore}(p) = \max_{\forall S, p \in S} \mathrm{CfdScore}(S)$. In this way, we can handle the difficulty of partial overlapping between segments and candidate windows. For the second difficulty, we divide each candidate window into $M = m \times m$ spatial bins and pool the confidence scores of the pixels in each bin. Because the classifiers are trained with the one-vs-all scheme, our spatial confidence map feature is class-specific. It leads to a $(M \times K)$-dimensional feature for each candidate window, where $K$ refers to the total number of object classes. After that, we encode it by additive kernels approximation mapping [25] and obtain the final feature representation of candidate windows. The feature generating process is illustrated in Figure 2. In the testing stage, we can concatenate this segment feature with the features from other object detection systems.

## 5 Experiments

In this section, we first evaluate the performance of e-SVM method on segment proposal classification, by using two new evaluation criterions for this task. After that, we apply our method to two essential tasks in computer vision: semantic segmentation and object detection. For semantic segmentation task, we test the proposed eSVM and latent eSVM on two different scenarios (i.e., with pixel-level groundtruth label annotation and with only bounding-box object annotation) respectively. For object detection task, we combine our confidence map feature with the state-of-the-art object detection system, and show our method can obtain non-trivial improvement on detection performance.

### 5.1 Performance evaluation on e-SVM

We use PASCAL VOC 2011 [9] segmentation dataset in this experiment. It is a subset of the whole PASCAL 2011 datasets with 1112 images in the training set and 1111 images in the validation set, with 20 foreground object classes in total. We use the official training set and validation set for training and testing respectively. Similar to [5], we extract 150 CPMC [4] segment proposals for each image and compute the second-order pooling features on each segment. Besides, we use the same sequential pasting scheme [5] as the inference algorithm in testing.

#### 5.1.1 Evaluation criteria

In literature [5], the supervised learning framework of segment-based prediction model either regressed the overlapping value or converted it to a binary classification problem via a threshold value, and evaluate the performance by certain task-specific criterion (i.e., the pixel-wise accuracy used for semantic segmentation). In this paper, we adopt a direct performance evaluation criteria for the segment-wise target class prediction task, which is consistent with the learning problem itself and not biased to particular tasks. Unfortunately, we have not found any work on this sort of direct performance evaluation, and thus introduce two new evaluation criteria for this purpose. We first briefly describe them as follows:

**Threshold Average Precision Curve (TAPC)** Although the ground-truth target value (i.e., the overlap rate of segment and bounding box) is a real value in the range of [0, 1], we can transform original prediction problem to a series of binary problems, each of which is conducted by thresholding the original groundtruth overlap rate. Thus, we calculate the Precison-Recall Curve as well as AP on each of binary classification problem, and compute the mean AP w.r.t. different threshold values as a performance measurement for the segment-based class confidence prediction problem.

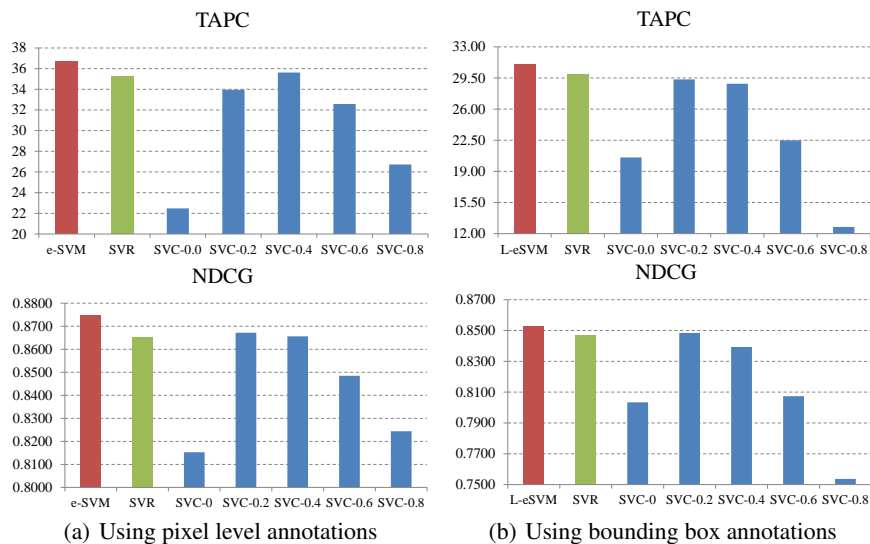

(a) Using pixel level annotations      (b) Using bounding box annotations

Figure 3: Performance evaluation and comparison to SVC and SVR

**Normalized Discounted Cumulative Gain (NDCG) [15]** Considering that a higher confidence value is expected to be predicted for the segment with higher overlap rate, we think this prediction problem can be treated as a ranking problem, and thus we use the Normalized Discounted Cumulative Gain (NDCG), which is common performance measurement for ranking problem, as another kind of performance evaluation criterion in this paper.

### 5.1.2 Comparisons to SVC and SVR

Based on the TAPC and NDCG introduced above, we evaluate the performance of our e-SVM model on PASCAL VOC 2011 segmentation dataset, and compare the results to two common methods (i.e. SVC and SVR) in literature. Note that we test the SVC's performance with a variety of binary classification problems, each of which are trained by using different threshold values (e.g., 0, 0.2, 0.4, 0.6 and 0.8 as shown in figure 3). In figure 3 (a) and (b), we show the experimental results w.r.t. the model/classifier trained with clean pixel-wise object class labels and weakly-labelled bounding-box annotation, respectively. For both cases, we can see that our method obtains consistently superior performance than SVC model for all different threshold values. Besides, we can see that the TAPC and NDCG of our method are higher than those of SVR, which is a popular regression model for continuously valued target variable based on the max-margin principle.

### 5.2 Results of semantic segmentation

For the semantic segmentation task, we test our e-SVM model with PASCAL VOC 2011 segmtation dataset using training set for training and validation set for testing. We evaluate the performance under two different data annotation settings, i.e., training with pixel-wise semantic class label maps and object bounding-box annotations. The accuracy w.r.t. these two settings are 36.8% and 27.7% respectively, which are comparable to the results of the state-of-the-art segment confidence prediction model (i.e., SVR) [5] used in semantic segmentation task.

### 5.3 Results of object detection

As mentioned in Section 4.2, one of the natural applications of our e-SVM method is the object detection task. Most recently, Girshick et.al [14] presented a Regions with CNN features method (RCNN) using the Convolutional Neural Network pre-trained on the ImageNet Dataset [6] and fine-tuned on the PASCAL VOC datasets. They achieved a significantly improvement over the previous state-of-the-art algorithms (e.g. Deformable Part-based Model (DPM) [11])and push the detection

|  | plane | bike | bird | boat | bottle | bus | car | cat | chair | cow |
|---|---|---|---|---|---|---|---|---|---|---|
| RCNN | **64.1** | 69.2 | 50.4 | 41.2 | 33.2 | 62.8 | 70.5 | 61.8 | 32.4 | 58.4 |
| Ours | 63.7 | **70.2** | **51.9** | **42.5** | **33.4** | **63.2** | **71.3** | **62.0** | **34.7** | **58.7** |
| Gain | -0.4 | 1.0 | 1.5 | 1.3 | 0.2 | 0.4 | 0.8 | 0.2 | 2.3 | 0.2 |
| RCNN (bb) | 68.1 | 72.8 | 56.8 | 43.0 | 36.8 | 66.3 | 74.2 | 67.6 | 34.4 | 63.5 |
| Ours (bb) | **70.4** | **74.2** | **59.1** | **44.7** | **38.0** | **67.2** | **74.6** | **69.0** | **36.7** | **64.3** |
| Gain | 2.3 | 1.4 | 2.3 | 1.6 | 1.2 | 1.0 | 0.3 | 1.3 | 2.3 | 0.8 |

|  | table | dog | horse | motor. | person | plant | sheep | sofa | train | tv | Average |
|---|---|---|---|---|---|---|---|---|---|---|---|
| RCNN | 45.8 | 55.8 | 61.0 | 66.8 | 53.9 | 30.9 | 53.3 | 49.2 | 56.9 | 64.1 | 54.1 |
| Ours | **47.8** | **57.9** | **61.2** | **67.5** | **54.9** | **34.5** | **55.8** | **51.0** | **58.4** | **65.0** | **55.3** |
| Gain | 2.0 | 2.1 | 0.3 | 0.8 | 1.0 | 3.7 | 2.5 | 1.8 | 1.6 | 0.9 | 1.2 |
| RCNN (bb) | 54.5 | 61.2 | 69.1 | 68.6 | 58.7 | 33.4 | 62.9 | 51.1 | 62.5 | 64.8 | 58.5 |
| Ours (bb) | **56.4** | **62.9** | **69.3** | **69.9** | **59.6** | **35.6** | **64.6** | **53.2** | **64.3** | **65.5** | **60.0** |
| Gain (bb) | 1.9 | 1.8 | 0.2 | 1.4 | 0.9 | 2.2 | 1.7 | 2.1 | 1.8 | 0.7 | 1.5 |

Table 1: Detection results on PASCAL 2007. "bb" means the result after applying bounding box regression. Gain means the improved AP of our system compared to RCNN under the same settings (both with bounding box or without). The better results in the comparisons are bold.

performance into a very high level (The average AP is 58.5 with boundary regularization on PASCAL VOC 2007).

A question arises: can we further improve their performance? The answer is yes. In our method, we first learn the latent e-SVM models based on the object bounding-box annotation, and calculate the spatial confidence map features as in section 4.2. Then we simply concatenate them with RCNN the features to train object classifiers on candidate windows. We use PASCAL VOC 2007 dataset in this experiment. As shown in table 1, our method can improve the average AP by 1.2 before applying bounding boxes regression. For some categories that the original RCNN does not perform well, such as potted plant, the gain of AP is up to 3.65. After applying bounding box regression for both RCNN and our algorithm, the gain of performance is 1.5 on average.

In the experiment, we set $m = 5$ and adopt average pooling on the pixel level confidence scores within each spatial bin. We also modified the bounding box regularization method used in [14] by augmenting the fifth layer features with additive kernels approximation methods [25]. It will lead to a slightly improved performance.

In summary, we achieved an average AP of 60.0, which is 1.5 higher than the best known results (the original RCNN with bounding box regression) of this dataset. Please note that we only use the annotations on PASCAL VOC 2007 to train the e-SVM classifiers and have not considered context. The results are expected to be further improved if the data in ImageNet is used.

## 6 Conclusion

We present a novel learning algorithm call e-SVM that can well handle the situation in which the labels of training data are continuous values whose range is a bounded interval. It can be applied to segment proposal classification task and can be easily extended to learn segment classifiers under weak supervision (e.g. only bounding box annotations are available). We apply this method on two major tasks of computer vision (i.e., semantic segmentation and object detection), and obtain the state-of-the-art object detection performance on PASCAL VOC 2007 dataset. We believe that, with the ever growing size of datesets, it is increasingly important to learn segment classifiers under weak supervision to reduce the amount of labeling required. In future work, we will consider using the bounding box annotation from large datasets, such as ImageNet, to further improve semantic segmentation performance on PASCAL VOC.

**Acknowledgements.** We gratefully acknowledge funding support from the National Science Foundation (NSF) with award CCF-1317376, and from the National Institute of Health NIH Grant 5R01EY022247-03. We also thank the NVIDIA Corporation for providing GPUs in our experiments.

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
