[Reviews · NeurIPS 2014]

Submitted by Assigned_Reviewer_6

The authors propose an expectation loss svm (e-svm) that can handle continuous attributes of positive instances in a learning problem and apply the model to learn visual segmentation classifiers under weak supervision. The methodology is validated for both semantic segmentation and object detection with promising results. The paper addresses an important problem of visual learning under weak supervision, the presentation is clear and well-motivated and the experiments are interesting and well executed. Although the technical content is an extension of the standard multiple instance learning framework (by replacing a binary with a soft weighting of positives), it applies in both the observed and the latent setting, and represents a valid contribution for NIPS.
Summary: An expectation loss svm method that can handle continuous attributes of positive instances in a learning problem and apply it to learn visual segmentation classifiers under weak supervision. Good motivation, clear presentation, relevant results for semantic segmentation and object detection.

Submitted by Assigned_Reviewer_30

The paper proposes a modification to the SVM learning algorithm that deals with weakly supervised data. Specifically, a per-example weight (between 0 and 1) is assigned to each training example. Then the loss function of the SVM is modified to take into account this weight. The method is then extended to the case when the per-example weight is not observed (i.e., is a latent variable).

The method is well described and the experimental results indicate that, for the semantic segmentation and object detection tasks, there may be value is using such an approach over treating each example with equal weight. However, I fail to see the difference between the proposed method and existing "example dependent costs", which are widely known and used in the machine learning community. SVMlight, for example, provides such a mechanism.

The extension to the case when the per-example weight is not observed is interesting but straightforward. In this case the problem is non-convex and no theoretical analysis is given. Indeed a strong regularization term is used suggesting that the per-example weights do not deviate far from their initial values. Can the authors comment?

Minor comment: The claim on L050 that "it is also not a standard regression problem since the positiveness belongs to a bounded interval [0, 1]". Logistic regression is very much standard and is bounded to [0, 1].

Minor spelling mistake in the abstract: "continues" -> "continuous"
Summary: The paper proposes a modified SVM learning algorithm in which the loss function is modified by a per-example weight. However, example dependent costs are already widely used in machine learning.

Submitted by Assigned_Reviewer_42

Summary:
The authors present a scheme for semantic labeling and object detection. The former is accomplished by generating candidate segments and extracting features for these; and then regressing on the Intersection-over-Union overlap ratio between a candidate segment and pixel-accurate ground truth. In the case of object detection, only weaker labels in terms of bounding boxes are available and the precise spatial extent is used as latent variable. Regression is performed using a weighted SVM and a weighted transductive SVM, respectively. Results are on par with the state of the art in semantic labeling, and the best currently known for object detection on VOC 2007.

Originality:
Algorithmic novelty is limited, and the authors unfortunately do not cite pertinent previous work. In particular, the weighted SVM is well-known (e.g. [R1,R2,R4] below, not cited). The authors put a sum constraint on the weights, which may be novel (I did not conduct an extensive literature search here). The "latent e-SVM" that the authors propose is a weighted version of a transductive SVM (e.g. [R3], not cited).

[R1] Suykens et al. Weighted least squares support vector machines: robustness and sparse approximation, Neurocomputing, 2002
[R2] X.Yang et al. Weighted Support Vector Machine for Data Classification, IJCNN, 2005
[R3] V. Vapnik, “Statistical Learning Theory”, 1998
[R4] M.Lapin et al, Learning Using Privileged Information: SVM+ and Weighted SVM, Neural Networks, 2014

Significance:
Outperforming the recent CNN results by a "conventional" learning approach is a feat and of great interest to the community.

Quality:
Crucial references to previous work are missing (see above). The object detection framework involves quite a bit of engineering, but not outside the norm. The default method for regression on the unit interval is (regularized) logistic regression, so these results should be reported as baseline to support the use of a sum-constrained weighted SVM. Authors should give more details on how they picked regularization coefficients \lambda_W and \lambda_R. The main strength of the paper are the good empirical results.

Clarity:
The title and abstract do not summarize the core contribution of the paper well. The abstract also insinuates that regression targets from the unit interval are somehow less informative than mere binary class memberships. This is not true, because the latter can always be obtained from the former by thresholding. For a NIPS audience, the paper spends rather too much space (up to and including page 4) on small variations of well-known algorithms, and passes somewhat quickly over the all-important feature engineering details (lines 254ff).

Minor comments
Line 19: continues => continuous
Line 94: boxes => box
Line 98: AP: Abbreviation not introduced
Line 350: SVM => SVC
Line 350: with every different values => for all threshold values

-----

I have upped my quality score by one, assuming the authors will not be defensive about the relation to previous work, but rather comment extensively on it and connections to it.
Summary: Great results, deficient references to earlier work, little algorithmic novelty. A paper that should certainly be published, possibly after some rewriting, and possibly rather at a computer vision conference.
Author Feedback
Author rebuttal: We thank the reviewers for their helpful comments. We will incorporate them in the revised manuscript.

(1) Reviewer 1: “…, I fail to see the difference between the proposed method and existing "example dependent costs", …”;
Reviewer 2: “… the authors unfortunately do not cite pertinent previous work. In particularly, the weighted SVM is well-known (e.g. [R1,R2,R4] below, not cited). The authors put a sum constraint on the weights, which may be novel ...”.

Reply: Reviewers 1 and 2 give comments about the difference between the proposed algorithm (i.e., e-SVM) and existing weighted SVMs (WSVM) or equivalently so-called example-dependent-cost methods. We think our e-SVM differs from previous WSVM methods in both motivation and formulation (loss function):

(i) In the WSVM literature, the goal of using example-dependent weights is to improve the robustness of model estimation [R1], alleviate the effect of outliers [R2], leverage privileged information [R4] or deal with unbalanced classification problems. However, our e-SVM is devoted to learning a prediction model for problems where only the “positiveness” of training examples is available and their class labels are unobserved. We apply our approach to two major and very difficult tasks (i.e., semantic segmentation and object detection) in computer vision, and obtain state-of-the-art performance. So the motivations and objectives of our method are different from previous WSVMs.

(ii) In terms of problem formulation, our e-SVM model is also different from existing WSVMs. For WSVMs there is only one loss term for each training example, which corresponds to its class label (either positive or negative). In our vision tasks, however, we can only obtain a continuous positiveness value instead of a binary label for each example, so the e-SVM minimizes the expectation of the hinge loss w.r.t. an underlying Bernoulli distribution of the class label, whose success probability is assumed to be a function of the observed positiveness. Hence, in e-SVM we simultaneously consider the costs of being a positive sample and a negative one for each example. This is a non-trivial difference and leads to a loss function for e-SVM which better utilizes fine-grained supervision information (i.e., the positiveness cue) from the training data.

Despite these differences, we agree there is a connection between our e-SVM and WSVMs. As mentioned in line 164~166, although its loss function is different from those of WSVMs, the e-SVM problem can be readily solved by a standard SVM solver (e.g., LibLinear or SVMlight) like those used for WSVMs. We think this is an advantage because it does not require a specific solver for the implementation of our e-SVM. As suggested by reviewer 2, we will cite these WSVM papers, and discuss the differences and connection to our work.

(2) Reviewer 1: “The extension to the case when the per-example weight is not observed is interesting but straightforward. In this case the problem is non-convex and no theoretical analysis is given.”

Reply: We briefly discussed the loss function and optimization algorithm of latent e-SVM in Sec. 3.2 (see line 182~186), and showed that the loss function L(w, u) is bi-convex. Hence, although L(w,u) is generally not a joint convex function over w and u, it is convex in each variable if the other one is fixed. Accordingly, the optimization algorithm we used (i.e., the alternate convex search), which is a common method for bi-convex problems, guarantees monotonic decrease of the objective function until convergence.

(3) Reviewer 1: “Indeed a strong regularization term is used suggesting that the per-example weights do not deviate far from their initial values. Can the authors comment?”

Reply: In general, the choice of regularization term is flexible in our latent e-SVM framework. We adopt the cross entropy form of Equ. (3) based on two reasons: (i) It is convex, which ensures the loss function is biconvex. (ii) It is applicable to our task (i.e., weakly-supervised training with bounding-box annotations). For most segment proposals, the overlap ratio w.r.t. the object bounding box (i.e., u^{bb}) is generally close to the ground-truth overlap ratio (i.e., the one w.r.t. pixel-wise label map), due to the fact that the bounding boxes tend to be very tight in object detection datasets (e.g., PASCAL VOC and ImageNet). Hence, it makes sense to use a relatively strong regularization term such that u will not be too far from u^{bb}. But at the same time, our algorithm is flexible and allows estimation of u for each individual segment proposal based on its e-SVM loss. So for some examples the values of u differ a lot from their initial values.

(4) Reviewer 2: “Authors should give more details on how they picked regularization coefficients \lambda_W and \lambda_R.”

Reply: The choice of values for \lambda_W and \lambda_R depends on specific task and dataset. Their values are determined via a validation set in training.

(5) Reviewer 2: The "latent e-SVM" that the authors propose is a weighted version of a transductive SVM (e.g. [R3], not cited).

Reply: We only use the training examples to learn the latent e-SVM model. On the contrast, the transductive learning incorporates the information from testing data. So our latent e-SVM is not a variant of transductive SVM.

(6) Reviewer 2: “The abstract also insinuates that regression targets from the unit interval are somehow less informative than mere binary class memberships. This is not true, because the latter can always be obtained from the former by thresholding.”

Reply: In abstract we did not indicate that the regression is less informative than the classification. Our e-SVM is different from the binary classification model (e.g. SVC), and performs better than the regression counterpart in our experiments.